# A viral protein promotes host SAMS1 activity and ethylene production for the benefit of virus infection

Shanshan Zhao[1†], Wei Hong[1,2†], Jianguo Wu[1,3], Yu Wang[1], Shaoyi Ji[1], Shuyi Zhu[1], Chunhong Wei[1], Jinsong Zhang[4], Yi Li[1]*

[1]State Key Laboratory of Protein and Plant Gene Research, College of Life Sciences, Peking University, Beijing, China; [2]The First Affiliated Hospital of Zhejiang Chinese Medical University, Hangzhou, China; [3]State Key Laboratory of Ecological Pest Control for Fujian and Taiwan Crops, Fujian Province Key Laboratory of Plant Virology, Institute of Plant Virology, Fujian Agriculture and Forestry University, Fuzhou, China; [4]State Key Lab of Plant Genomics, Institute of Genetics and Developmental Biology, Chinese Academy of Sciences, Beijing, China

*For correspondence:
liyi@pku.edu.cn

[†]These authors contributed equally to this work

Competing interests: The authors declare that no competing interests exist.

**Abstract** Ethylene plays critical roles in plant development and biotic stress response, but the mechanism of ethylene in host antiviral response remains unclear. Here, we report that *Rice dwarf virus* (RDV) triggers ethylene production by stimulating the activity of S-adenosyl-L-methionine synthetase (SAMS), a key component of the ethylene synthesis pathway, resulting in elevated susceptibility to RDV. RDV-encoded Pns11 protein specifically interacted with OsSAMS1 to enhance its enzymatic activity, leading to higher ethylene levels in both RDV-infected and Pns11-overexpressing rice. Consistent with a counter-defense role for ethylene, Pns11-overexpressing rice, as well as those overexpressing *OsSAMS1*, were substantially more susceptible to RDV infection, and a similar effect was observed in rice plants treated with an ethylene precursor. Conversely, *OsSAMS1*-knockout mutants, as well as an *osein2* mutant defective in ethylene signaling, resisted RDV infection more robustly. Our findings uncover a novel mechanism which RDV manipulates ethylene biosynthesis in the host plants to achieve efficient infection.
DOI: https://doi.org/10.7554/eLife.27529.001

## Introduction

Rice is a staple food crop in many regions, a model monocot plant for research, and a host to many viruses (*Wu et al., 2015*; *2017*). Viral infection causes substantial losses in yield and quality in rice crops and current knowledge on the antiviral responses of monocotyledonous crops is very limited (*Soosaar et al., 2005*; *Mandadi and Scholthof, 2013*; *Wang, 2015*). *Rice dwarf virus* (RDV), a member of the genus *Phytoreovirus* in the family *Reoviridae*, is one of the most widespread and devastating viruses that infect rice (*Wu et al., 2015*; *Jin et al., 2016*). RDV is transovarially transmitted by the green rice leafhopper (*Nephotettix cincticeps*) in a persistent-propagative manner (*Cao et al., 2005*; *Zhou et al., 2007*; *Wei and Li, 2016*). RDV infection greatly inhibits rice growth and causes severe symptoms including dwarfism, increased tillering, and white chlorotic specks and dark-green discoloration on the leaves. RDV has a double-stranded RNA genome consisting of 12 segments (*S1* to *S12*). Seven segments, *S1*, *S2*, *S3*, *S5*, *S7*, *S8*, and *S9*, encode structural proteins P1, P2, P3, P5, P7, P8, and P9, respectively, which form double-layered virions; the remaining segments, *S4*, *S6*, *S10*, *S11*, and *S12,* encode the nonstructural proteins Pns4, Pns6, Pns10, Pns11, and Pns12 (*Cao et al., 2005*; *Zhou et al., 2007*; *Liu et al., 2014*; *Jin et al., 2016*; *Wei and Li, 2016*).

**eLife digest** Rice provides food for billions of people all over the world, but diseases caused by plant-infecting viruses cause serious risks to the production of rice. As a result, there is an urgent demand for developing new and impactful ways to help defend rice plants from harmful viruses. Toward this goal, it will be important to better understand how viruses actually cause diseases in plants.

Plants make chemicals known as hormones to control their own development, and hormone production is often disturbed when viruses infect rice plants. Many viruses cause infected plants to make more of a gaseous hormone called ethylene, which benefits the viruses. Yet, it is still not known how virus infection induces the production of more ethylene.

Zhao, Hong et al. have exposed rice plants to infection with a virus called rice dwarf virus. Infected plants made more ethylene than normal, which did indeed help the virus to infect. Further experiments then showed that an enzyme that makes one of the building blocks needed to produce ethylene became more active after infection with this virus. Next, Zhao, Hong et al. engineered rice plants to make more or less of this building block – which is known as S-adenosyl-L-methionine or SAM for short. Plants with too much SAM were less able to defend themselves against the virus, while plants that lacked SAM were better able to fight off viral infection.

Zhao, Hong et al. suggest that engineering rice plants to make less of the SAM-producing enzyme could make them more resistant to viruses. Further work will also be needed to find out why ethylene benefits viral infection, and to confirm whether ethylene also performs similar roles when rice is infected with other viruses.

DOI: https://doi.org/10.7554/eLife.27529.002

To survive under the continuous threat of viral infection, plants have evolved multiple defense mechanisms that are activated via different signal transduction pathways. The intensively studied pathways, based on dicot model plants, include nucleotide-binding-site leucine-rich-repeat dominant resistance genes (*R*-genes), recessive resistance genes (eukaryotic initiation factors (eIFs)), RNA interference (RNAi) antiviral immunity, and phytohormone-mediated resistance pathways; these pathways interact synergistically or antagonistically and result in a highly complex three-dimensional defense signaling network (*Kasschau et al., 2003*; *Baulcombe, 2004*; *Soosaar et al., 2005*; *Ding, 2010*; *Endres et al., 2010*; *Mandadi and Scholthof, 2013*; *Nicaise, 2014*; *Carbonell and Carrington, 2015*; *Collum and Culver, 2016*; *Wang, 2015*; *Wu et al., 2017*). As a counter-defense, plant viruses often manipulate plant responses for their own benefit. For example, viruses have evolved strategies to target hormone pathways, often exploiting the antagonistic interactions mediated by phytohormones such as salicylic acid (SA), jasmonic acid (JA), and ethylene (*Soosaar et al., 2005*; *Broekaert et al., 2006*; *Pieterse et al., 2009*; *Santner et al., 2009*; *Denancé et al., 2013*; *Mandadi and Scholthof, 2013*; *Alazem and Lin, 2015*).

The gaseous phytohormone ethylene functions in seed germination and organ senescence, as well as in the response of plants to abiotic and biotic stresses (*Broekaert et al., 2006*; *van Loon et al., 2006*; *Alazem and Lin, 2015*; *Kazan, 2015*). However, the functions of ethylene in the plant response to viral infection remain poorly understood. Previous studies found that ethylene could modulate host defense in both positive and negative manners (*Knoester et al., 2001*; *Love et al., 2007*; *Pieterse et al., 2009*; *Santner et al., 2009*; *Chen et al., 2013b*; *Zhu et al., 2014*; *Casteel et al., 2015*). The P6 protein encoded by *Cauliflower mosaic virus* (CaMV) was found to interact with components of the ethylene-signaling pathway, and transgenic *Arabidopsis* expressing P6 became less responsive to ethylene treatment and more resistant to CaMV infection (*Geri et al., 2004*). Another study used *ein2* (*ethylene insensitive 2*) and *etr1* (*ethylene response 1*) mutants and found that the ethylene-signaling pathway is required for *Turnip mosaic virus* (TuMV)-mediated suppression of resistance to the green aphid, *Myzus persicae,* in *Arabidopsis,* and that TuMV may modulate ethylene responses to increase plant susceptibility to viral infection (*Casteel et al., 2015*). *Chen et al. (2013b)* reported that *Arabidopsis* plants with mutations of the ethylene biosynthesis pathway, such as *acs1* (*1-aminocyclopropane-1-carboxylate synthase*), *erf106* (*ethylene responsive transcription factor 106*), and *ein2*, were resistant to *Tobacco mosaic virus* (TMVcg). Exogenous

application of 1-aminocyclopropane-1-carboxylic acid (ACC, a precursor in the ethylene biosynthesis pathway) enhanced TMVcg accumulation in the infected plants (*Chen et al., 2013a*). By contrast, ethylene signaling was shown to be essential for systemic resistance to *Chilli veinal mottle virus* in tobacco (*Zhu et al., 2014*). Thus, the molecular mechanisms by which ethylene affects host defenses and counter-defenses remain unclear.

In plants, S-adenosyl-L-methionine synthetase (SAMS) [EC 2.5.1.6] catalyzes the conversion of L-methionine (L-Met) and ATP into S-adenosyl-L-methionine (AdoMet, SAM), which serves as a precursor of ethylene and polyamines. The SAMS enzyme is induced by biotic and abiotic stress and is involved in the regulation of development through the histone and DNA methylation pathway (*Roje, 2006*; *Li et al., 2011*; *Chen et al., 2013a*; *Gong et al., 2014*; *Yang et al., 2015*). A previous study demonstrated that RDV infection perturbed the expression of several ethylene-response genes such as ERFs (Ethylene Response Factors) (*Satoh et al., 2011*; *Abiri et al., 2017*), indicating that ethylene is involved in the interaction between RDV and rice. However, it is unclear how the ethylene biosynthesis and signaling pathway functions in this interaction.

In the current study, we report that the RDV-encoded non-structural protein Pns11 enhances rice susceptibility to RDV by interacting with OsSAMS1, enhancing its enzymatic activity and leading to increasing production of SAM, ACC, and ethylene. As SAMS and ethylene are key regulators of many biological processes, the capability of RDV-encoded Pns11 to interact specifically with OsSAMS1 and to regulate the ethylene biosynthesis and signaling pathway may represent a novel mechanism by which RDV maximizes its own infection. This study provides a novel mechanism through which ethylene biosynthesis and signaling respond to viral infection. These findings significantly broaden our knowledge of virus–host interactions and provide novel targets for engineered resistance to viruses.

## Results

### Overexpression of Pns11 in rice enhances susceptibility to RDV infection

RDV-encoded Pns11 protein was previously found to function as a component of viroplasms (*Wei et al., 2006*). To investigate whether Pns11 plays an important role in RDV infection, transgenes encoding Pns11 were introduced into the rice cultivar Zhonghua 11 to generate Pns11-overexpression plants, referred to as *S11* OX lines hereafter. Three transgenic lines #3, #5, and #11 were chosen for detailed analysis based on the detection of both Pns11 mRNA and the HA-tagged protein (*Figure 1—figure supplement 1A,B*). No obvious differences in phenotype were observed between the *S11* OX lines and wild-type (WT) rice except for grain size (*Figure 1—figure supplement 1C–F*). We then inoculated 30 seedlings (14-d-old) from each *S11* OX line with RDV using viruliferous leafhoppers (*Supplementary file 1A*) and observed disease symptoms. At four weeks post inoculation (4 wpi), the *S11* OX lines exhibited more severe RDV infection symptoms with more stunting and chlorotic flecks on the leaves than the WT control plants (*Figure 1A*). Three RDV RNA genome segments and their encoded proteins were evaluated by northern and western blot assays. The results showed increased accumulation in two of the *S11* OX lines (except *S11* OX#3) relative to the WT plants (*Figure 1B,C*). In addition, the infection rates were higher in the *S11* OX#5 and *S11* OX#11 lines than in the the WT (*Figure 1D*, *Supplementary file 2A*). Taken together, these results showed that overexpression of Pns11 compromised rice defense to RDV.

### Pns11 specifically interacts with OsSAMS1 to enhance OsSAMS1 enzymatic activity

The results described above showed that Pns11 overexpression increases rice susceptibility to virus infection. To elucidate the mechanism behind this, we tried to identify rice factors that interact with Pns11 by conducting a yeast two-hybrid screen of a rice cDNA library, with RDV-encoded Pns11 as the bait. This screen identified OsSAMS1 as a strong interaction partner of Pns11. The rice genome encodes three members of the predicted SAMS family, OsSAMS1, OsSAMS2, and OsSAMS3, which show high levels of sequence identity in their DNA and deduced amino acid sequences (*Li et al., 2011*). However, Pns11 only interacted with OsSAMS1, and not with OsSAMS2 or OsSAMS3 in

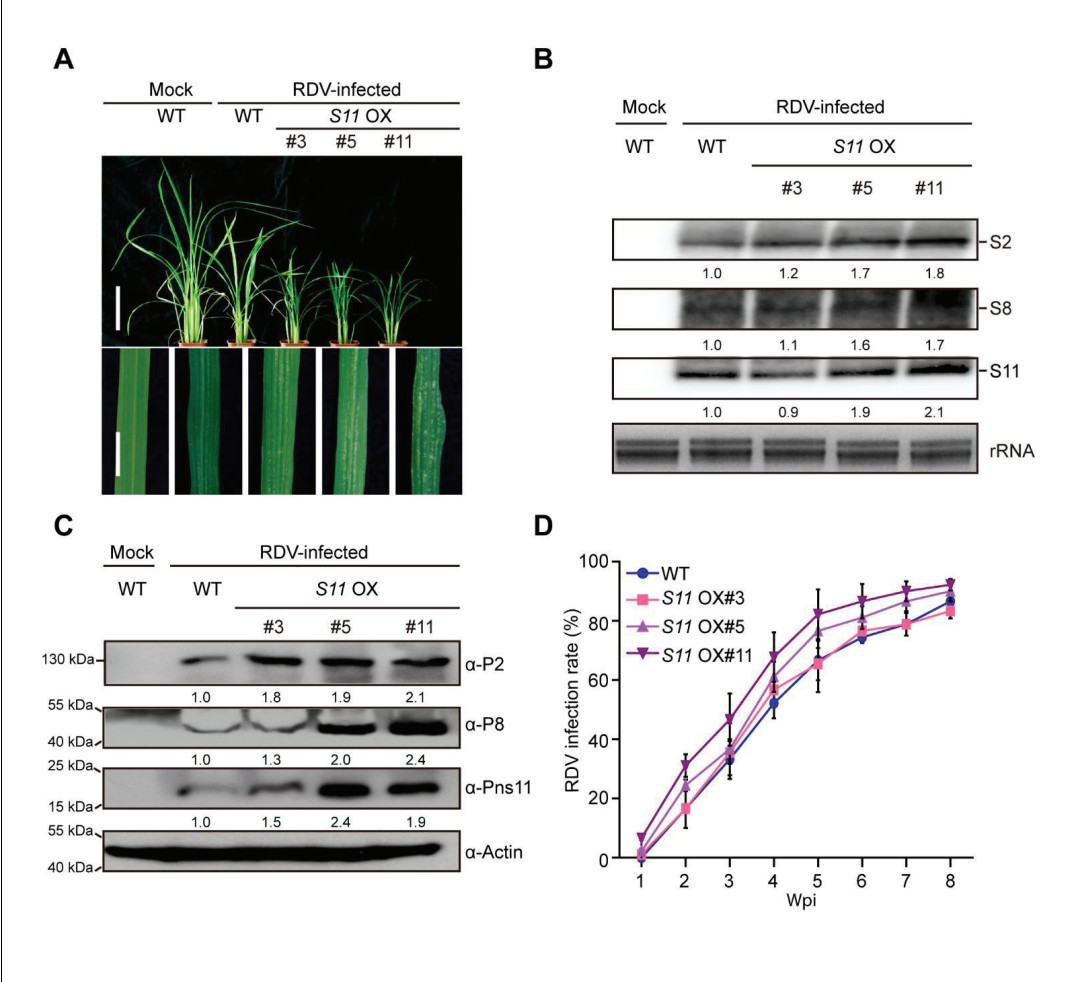

**Figure 1.** Pns11 overexpression lines are more susceptible to RDV infection than WT plants. (**A**) Symptoms of the mock-inoculated WT or RDV-infected WT (non-transgenic) plants and *S11* OX transgenic plants; images were taken at 4 wpi. Scale bars = 10 cm (upper panel) and 5 cm (lower panel). (**B**) Detection of RDV *S2*, *S8*, and *S11* genomic segments in the mock-inoculated WT or RDV-infected WT (non-transgenic) plants and in *S11* OX transgenic plants by northern blot. The blots were hybridized with radiolabeled riboprobes specific for each RNA segment. rRNAs were stained with ethidium bromide and served as loading controls. Tissues were collected at 4 wpi. (**C**) Detection of RDV P2, P8, and Pns11 protein in the mock-inoculated WT or RDV-infected WT (non-transgenic) plants and *S11* OX transgenic plants by western blot. Actin was probed and served as a loading control. Tissues were collected at 4 wpi. (**D**) The incidences of infection, which were determined by visual assessment of disease symptoms of 30 individual plants for each case at 0–8 wpi. Means and standard deviations were obtained from three independent experiments.

DOI: https://doi.org/10.7554/eLife.27529.003

The following figure supplement is available for figure 1:

**Figure supplement 1.** Phenotype and identification of *S11*-overexpressing lines.

DOI: https://doi.org/10.7554/eLife.27529.004

yeast. Moreover, OsSAMS1 specifically interacted with Pns11, but not with other RDV-encoded proteins (*Figure 2A*, *Figure 2—figure supplement 1A*).

To further test this specific interaction in plant cells, we performed a co-IP experiment by co-expressing hemagglutinin (HA)-epitope-tagged Pns11 and FLAG-tagged OsSAMS1, OsSAMS2, or OsSAMS3 in a transient expression assay in *Nicotiana benthamiana* leaves, followed by immunoprecipitation with FLAG-tag antibodies and HA-tag antibodies (*Figure 2B*, *Figure 2—figure supplement 1B,C*). This set of experiments confirmed the highly specific interaction between Pns11 and OsSAMS1. We further verified this interaction using a firefly luciferase (LUC) complementation imaging assay. Constructs encoding Pns11 fused with the N-terminus of LUC (Pns11-nLUC) and the C-terminus of LUC fused with OsSAMS1 (cLUC-OsSAMS1) were co-infiltrated into *N. benthamiana* leaves

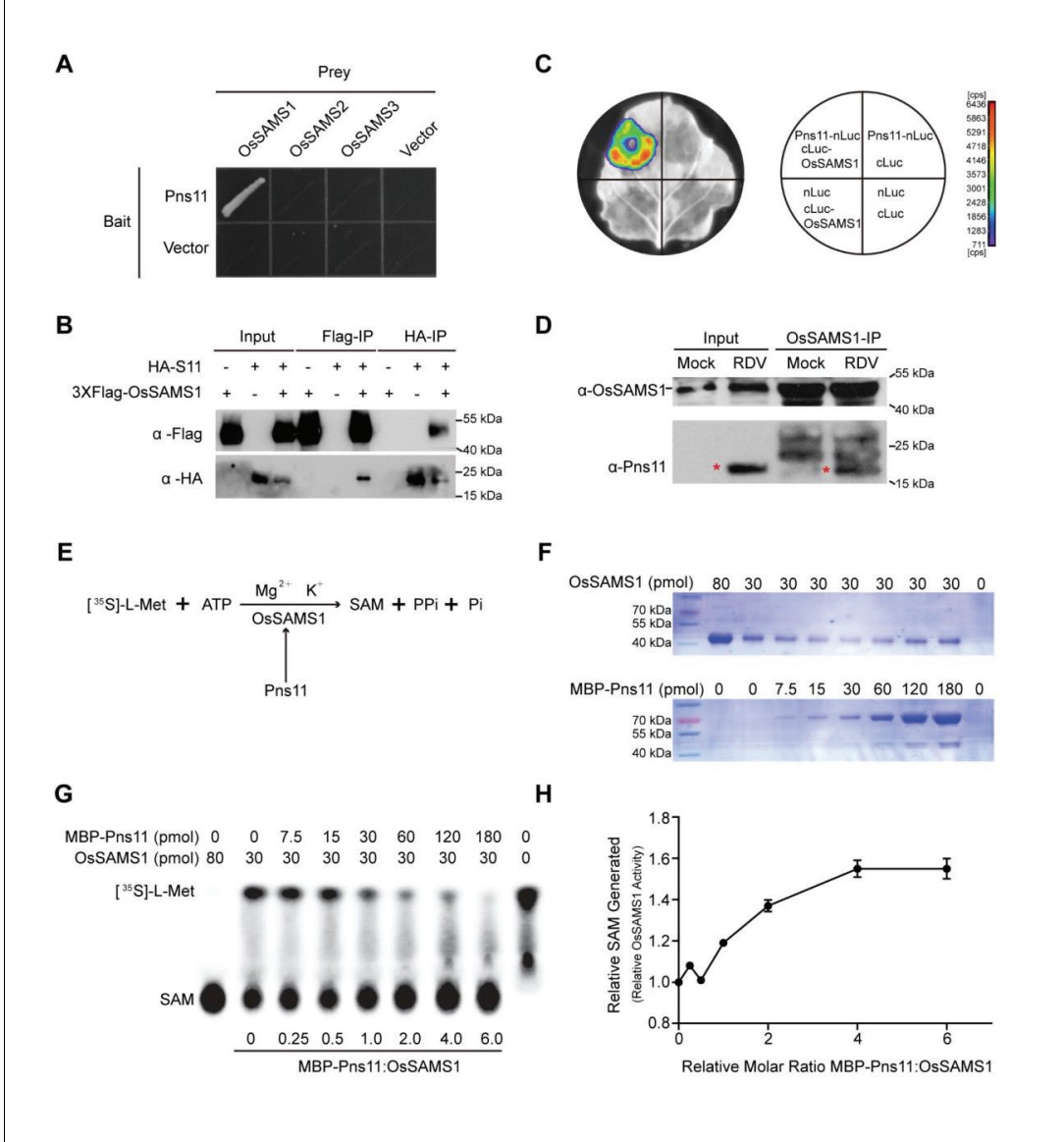

**Figure 2.** Pns11 interacts with OsSAMS1 and enhances its activity. (**A**) Yeast two-hybrid assay for the interaction of Pns11 with OsSAMS1. The bait vector contained full-length Pns11; the prey vector contained OsSAMS1, OsSAMS2, or OsSAMS3. Yeast strains were cultured on the Trp –Leu –His –Ade selection medium. (**B**) Co-IP assay for the interaction of Pns11 with OsSAMS1. Pns11 and OsSAMS1 proteins were transiently expressed in *Nicotiana benthamiana* leaves for 3 days. Plant extracts were then immunoprecipitated using anti-Flag or anti-HA antibodies, separated on a 10% SDS-PAGE gel, and blotted with anti-Flag or anti-HA antibodies. (**C**) Luciferase complementation imaging assay for the interaction of Pns11 and OsSAMS1. *Agrobacterium* strain EHA105 harboring different construct combinations was infiltrated into different *N. benthamiana* leaf regions. After 3 days of infiltration, luciferase activities were recorded in these regions. Cps, signal counts per second. (**D**) In *vivo* pull-down assay confirmed the interaction between Pns11 and OsSAMS1 during RDV infection in rice using α-OsSAMS1 antibody. The red asterisks indicate the location of Pns11. Tissues were collected at 4 wpi. (**E**) Diagram of the assay. In the reaction, OsSAMS1 catalyzes a two-step reaction in the presence of $Mg^{2+}$ and $K^+$ that involves the transfer of the adenosyl moiety of ATP to methionine to form SAM and tripolyphosphate, which is subsequently cleaved to PPi and Pi. Conversion of L-[$^{35}$S]-Met to SAM is activated by Pns11. (**F**) Coomassie brilliant blue staining of OsSAMS1 and Pns11 at the varying amounts of used in this assay. (**G**) Autoradiograph of a representative chromatogram showing SAM generated by OsSAMS1 in reactions containing varying molar ratios of maltose-binding protein (MBP)-Pns11 to OsSAMS1. The positions of labeled L-[$^{35}$S]-Met substrate and SAM product are indicated. L-[$^{35}$S]-Met and SAM in each reaction was calculated after phosphorimager quantitation of radioactivity in individual spots. (**H**) Stoichiometry of activation. The graph illustrates relative OsSAMS1 activity with increasing molar ratios of MBP-Pns11:OsSAMS1. Data were obtained from three independent experiments.

DOI: https://doi.org/10.7554/eLife.27529.005

The following figure supplements are available for figure 2:

**Figure supplement 1.** RDV Pns11 specifically interacts with OsSAMS1 and does not affect OsSAMS1 expression.

*Figure 2 continued on next page*

*Figure 2 continued*

DOI: https://doi.org/10.7554/eLife.27529.006

**Figure supplement 2.** Pns11 and OsSAMS1 are co-localized in both nucleus and cytoplasm.

DOI: https://doi.org/10.7554/eLife.27529.007

**Figure supplement 3.** Neither GFP nor P9 affects OsSAMS1 activity in *vitro*.

DOI: https://doi.org/10.7554/eLife.27529.008

**Figure supplement 4.** Pns11 does not affect OsSAMS2 activity in *vitro*.

DOI: https://doi.org/10.7554/eLife.27529.009

**Figure supplement 5.** Pns11 does not affect the affinity of OsSAMS1 to the substrates L-Met or ATP.

DOI: https://doi.org/10.7554/eLife.27529.010

**Figure supplement 6.** Multiple alignments of SAM synthetase proteins.

DOI: https://doi.org/10.7554/eLife.27529.011

**Figure supplement 7.** Gel-filtration analysis of Pns11 and OsSAMS1.

DOI: https://doi.org/10.7554/eLife.27529.012

for transient co-expression of these two fusion proteins. A luminescence signal was only detected in Pns11-nLUC/cLUC-OsSAMS1 co-expression regions but not in the negative controls (*Figure 2C*).

Finally, we also performed a in *vivo* pull-down assay with whole-cell lysates from non-infected controls and RDV-infected rice plants and found that Pns11 and OsSAMS1 interacted in RDV-infected rice cells (*Figure 2D*). Bimolecular fluorescence complementation (BiFC) analysis also demonstrated that Pns11 and OsSAMS1 interacted and were co-localized in both nucleus and cytoplasm (*Figure 2—figure supplement 2*). Taken together, our data strongly suggest that Pns11 specifically interacts with OsSAMS1 in *vitro* and in *vivo*.

To evaluate the biological significance of this specific interaction, we tested the level of OsSAMS1 in *S11* OX lines. We used *S11* OX rice tissues at three stages (5-leaf, 6-leaf, and 10-leaf stage) for real-time PCR (qRT-PCR) measurements and western blot. The *OsSAMS1* mRNA and OsSAMS1 protein level did not change in response to Pns11 (*Figure 2—figure supplement 1D,E*). We then designed an assay to detect the enzymatic activity of OsSAMS1 in *vitro*. SAMS catalyzes the two-step reaction that produces SAM, pyrophosphate (PPi), and orthophosphate (Pi) from ATP and L-Met (*Figure 2E*). Pns11 fused to maltose-binding protein (MBP) (MBP-Pns11) and OsSAMS1 fused to glutathione S-transferase (GST-OsSAMS1) were expressed in *Escherichia coli* BL21 cells and partially purified. To rule out the effect of the tags, the GST tag was cleaved to obtain pure OsSAMS1. For unknown reasons, the yield of Pns11 was extremely low if the MBP tag was removed. Therefore, we used MBP-GFP and MBP-P9 (a structural protein of RDV), which did not interact with OsSAMS1, as negative controls; we also used OsSAMS2, which did not interact with Pns11, as another negative control. The addition of L-[$^{35}$S]-Met mimics the natural substrate and allowed us to quantify the enzymatic activity of OsSAMS1 by measuring the amount of labeled SAM produced. OsSAMS1 was pre-incubated for 20 min with varying amounts of Pns11 (no Pns11 to a six-fold molar excess of Pns11: OsSAMS1) (*Figure 2F*). Solutions containing ATP, L-[$^{35}$S]-Met, KCl, and MgCl$_2$ were then added to the reaction mixtures and the reactions were allowed to proceed for another 20 min at 30°C. The reaction was blocked by the addition of EDTA. A reaction with excess OsSAMS1 and no Pns11 was used to label the location of the SAM, another reaction lacking OsSAMS1 and Pns11 was used to label the location of free L-[$^{35}$S]-Met. Production of SAM and the remaining L-[$^{35}$S]-Met were monitored by thin-layer chromatography (*Figure 2G*). The enzymatic activity of OsSAMS1 was enhanced by nearly 60% at a 6:1 molar ratio of Pns11:OsSAMS1 (*Figure 2H*). In the two negative control reactions with the same molar ratio of Pns11:OsSAMS1, neither P9 nor GFP affected OsSAMS1 activity (*Figure 2—figure supplement 3*). In addition, Pns11 did not affect OsSAMS2 enzymatic activity (*Figure 2—figure supplement 4*). The results described above demonstrated that Pns11 only interacts with OsSAMS1 and enhances the activity of OsSAMS1 for SAM synthesis in *vitro*.

## SAM, ACC, and ethylene contents increased in *S11*- and *OsSAMS1*-overexpression lines and decreased in *OsSAMS1* CRISPR/Cas9 knockout and RNAi lines

The results described above showed that Pns11 specifically interacts with OsSAMS1 and enhances its enzymatic activity to increase SAM production in *vitro*. SAM serves as the precursor of polyamine

and ethylene, and a previous study showed that most ethylene-response genes, such as ERFs and PRs (Pathogenesis-related genes) are regulated in RDV-infected rice (*Satoh et al., 2011*; *Abiri et al., 2017*; *Agrawal et al., 2001*), indicating an important role of ethylene in RDV infection. Thus we wondered whether overexpression of Pns11 in rice would enhance the enzymatic activity of OsSAMS1 and promote the synthesis of SAM, ACC, and ethylene in *vivo*. *S11* OX lines were used for analysis and the results showed that SAM, ACC and ethylene contents increased in two of the *S11* OX lines (but not in *S11* OX#3) (*Figure 3A–C*). To further confirm whether ethylene levels were affected by changes in SAM levels, *OsSAMS1* was introduced into the rice Zhonghua 11 background to generate *OsSAMS1*-overexpression (OX) lines. We also generated *OsSAMS1* RNAi lines (knockdown) and *OsSAMS1* knockout (KO) lines using CRISPR/Cas9. Positive transgenic rice lines were obtained through antibiotic selection and molecular screening. Among the *OsSAMS1*-overexpression lines, three (*OsSAMS1* OX#10, OX#17, and OX#25) were further analyzed. RNAi lines were characterized and classified as strong (RNAi-S) or weak (RNAi-W) according to the level of downregulation of *OsSAMS1* (*Figure 3—figure supplement 1A–C*). Two independent *Ossams1* KO rice lines (KO#31 and KO#39) with a mutation at different codons in the coding sequence were obtained (*Figure 3—figure supplement 1D*). Seed germination was suppressed in the *OsSAMS1* RNAi and

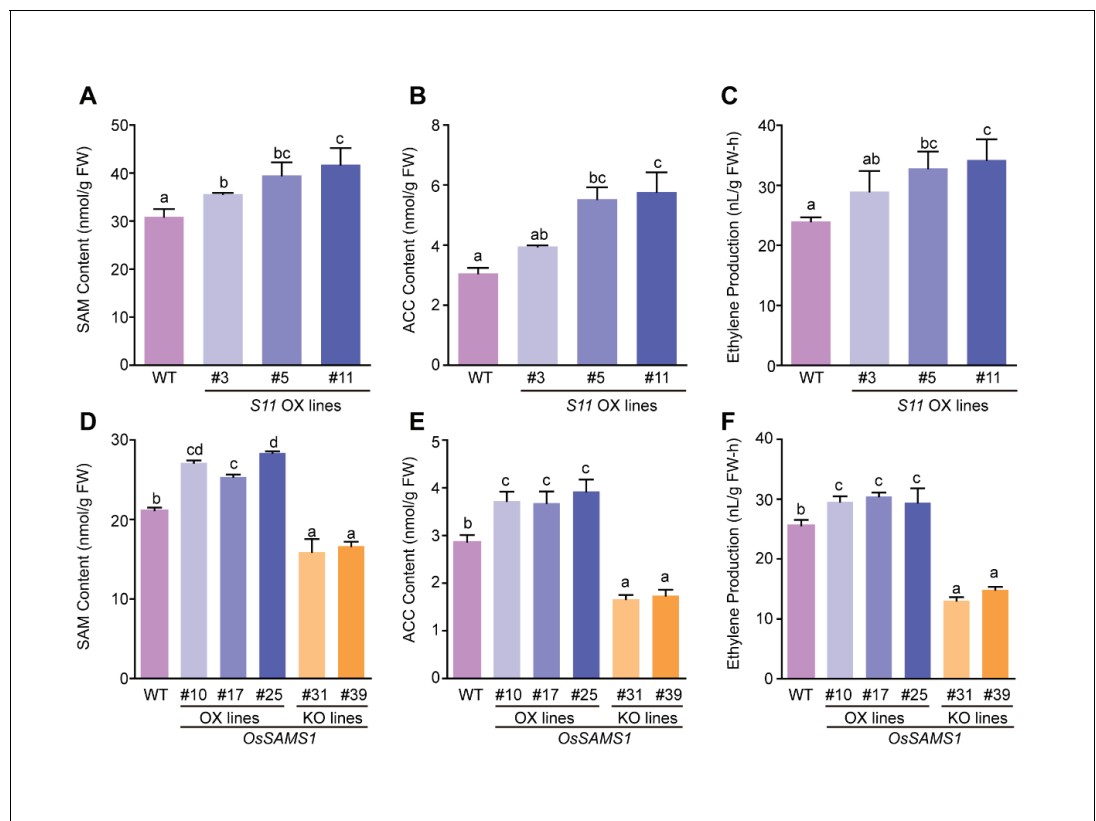

**Figure 3.** SAM, ACC, and ethylene contents in *OsSAMS1* and *S11* transgenic lines. (A) SAM contents in the *S11*-overexpression lines and WT plants (using 40-d-old seedlings). (B) ACC contents in the *S11*-overexpression lines and WT plants (using 40-d-old seedlings). (C) Ethylene contents in the *S11*-overexpression lines and WT plants (using 40-d-old seedlings). (D) SAM contents in the *OsSAMS1* transgenic lines and WT plants (using 40-d-old seedlings). (E) ACC contents in the *OsSAMS1* transgenic lines and WT plants (using 40-d-old seedlings). (F) Ethylene contents in the *OsSAMS1* transgenic lines and WT plants (using 40-d-old seedlings). Tukey's honestly significant difference post hoc test was performed for multiple comparisons. Letters indicate significant differences, p<0.05. Data are from three replicates. FW, fresh weight.

DOI: https://doi.org/10.7554/eLife.27529.013

The following figure supplements are available for figure 3:

**Figure supplement 1.** Phenotype and identification of *OsSAMS1* transgenic lines.
DOI: https://doi.org/10.7554/eLife.27529.014

**Figure supplement 2.** SAM, ACC and ethylene contents in *OsSAMS1* transgenic lines.
DOI: https://doi.org/10.7554/eLife.27529.015

KO lines and the suppression could be rescued by supplementation with SAM and ethylene (*Figure 3—figure supplement 1E*). The RNAi and KO lines also showed developmental defects, including dwarfism and reduced fertility (*Figure 3—figure supplement 1F*) (*Li et al., 2011*). Previous studies have demonstrated that ACC and ethylene contents increased in *OsSAMS1* OX lines and decreased in *OsSAMS1* RNAi transgenic lines, relative to WT plants (*Chen et al., 2013b*). In our study, SAM, ACC, and ethylene contents all increased in the *OsSAMS1* OX lines and decreased in the *OsSAMS1* RNAi and KO lines (*Figure 3D–F*, *Figure 3—figure supplement 2*). These results further demonstrated that Pns11 enhances the enzymatic activity of OsSAMS1 and alters the expression of SAMS in *vivo*, resulting in a corresponding change in the production of ACC and ethylene.

## Increases in SAM, ACC, and ethylene contents decrease rice tolerance to virus infection

Overexpression of *OsSAMS1* resulted in increased levels of SAM, ACC, and ethylene, and knockout of *OsSAMS1* resulted in decreased levels of SAM, ACC, and ethylene in rice. To investigate whether RDV infection, virus accumulation, and the host response is affected by SAM, ACC, and ethylene contents in the *OsSAMS1* OX, RNAi, and KO lines, we inoculated 30 seedlings (14-d-old) from each line (WT, OX#10, OX#17, OX#25, RNAi-S, RNAi-W, KO#31, and KO#39) with RDV using viruliferous leafhoppers and observed the resulting disease symptoms (*Supplementary file 1B,C*). At 4 wpi, all three OX lines displayed more severe stunting symptoms and chlorotic flecks at the infection site than the WT control plants, suggesting that they were more susceptible to RDV infection, while the RNAi and KO lines showed greater tolerance (*Figure 4A*, *Figure 4—figure supplement 1A*). Northern and western blot assays revealed that RDV accumulation was higher in the OX lines than in the WT, but much lower in the RNAi and KO lines (*Figure 4B,C*, *Figure 4—figure supplement 1B,C*). The infection rate was also higher in the OX lines, but much lower in the RNAi and KO lines compared to the WT plants (*Figure 4D*, *Figure 4—figure supplement 1D*, *Supplementary file 2B, C*). Taken together, these results suggested that overexpression of *OsSAMS1* enhances RDV infection whereas knockout of *OsSAMS1* reduces RDV infection, indicating a positive role of ethylene in RDV infection.

## Blocking ethylene signaling significantly enhances host tolerance to viral infection

Our results demonstrated that the endogenous accumulation of ethylene negatively regulates the plant antiviral defense response to RDV infection. However, it is not clear whether ethylene signaling is involved in the response of rice to RDV infection. The rice *MHZ7* gene (named *OsEIN2*), which encodes a membrane protein homologous to EIN2, a central component of ethylene signaling in *Arabidopsis*, also plays a key role in the rice ethylene signaling pathway (*Ma et al., 2013*; *Li et al., 2015*). The ethylene signaling mutant *mhz7* (*osein2*) is insensitive to ethylene in both the root and coleoptile. To investigate whether blocking the ethylene signaling pathway affects the rice antiviral defense response, we inoculated the *osein2* mutant and two *OsEIN2*-overexpression lines (OX#2 and OX#3) with RDV using viruliferous leafhoppers and observed the resulting disease symptoms (*Supplementary file 1D*). At 4 wpi, the *OsEIN2* OX#2 and OX#3 lines showed enhanced susceptibility with more severe stunting and chlorotic flecks on the leaves than did the WT control plants. By contrast, the *osein2* mutant showed much milder dwarfism and fewer chlorotic flecks on the leaves (*Figure 5A*). Northern and western blot assays indicated that RDV accumulation was much higher in the *OsEIN2* OX lines than in the WT plants, but lower in the mutant lines (*Figure 5B,C*). RDV infection rates among the *OsEIN2* OX lines, the WT, and the mutant lines diminished with time following infection. At 8 wpi, the infection rate of the *osein2* mutant was only 43%, which was significantly lower than that of the WT (84%) and the *OsEIN2* OX lines (OX#2, 96%; OX#3, 99%) (*Figure 5D*, *Supplementary file 2D*). These results suggested that the ethylene-response mutation enhances the rice antiviral defense response and that overexpression of *OsEIN2* increases host susceptibility. This is consistent with above results.

To further confirm that antiviral response was conferred through the ethylene signaling pathway, we overexpressed *OsSAMS1* in *osein2* mutant background rice and obtained three positive transgenic lines J119#1, J119#2 and J119#3 (*Figure 5—figure supplement 1*) for further analysis and RDV infection assay. Four weeks post inoculation, we found that the J119 lines, as well as the

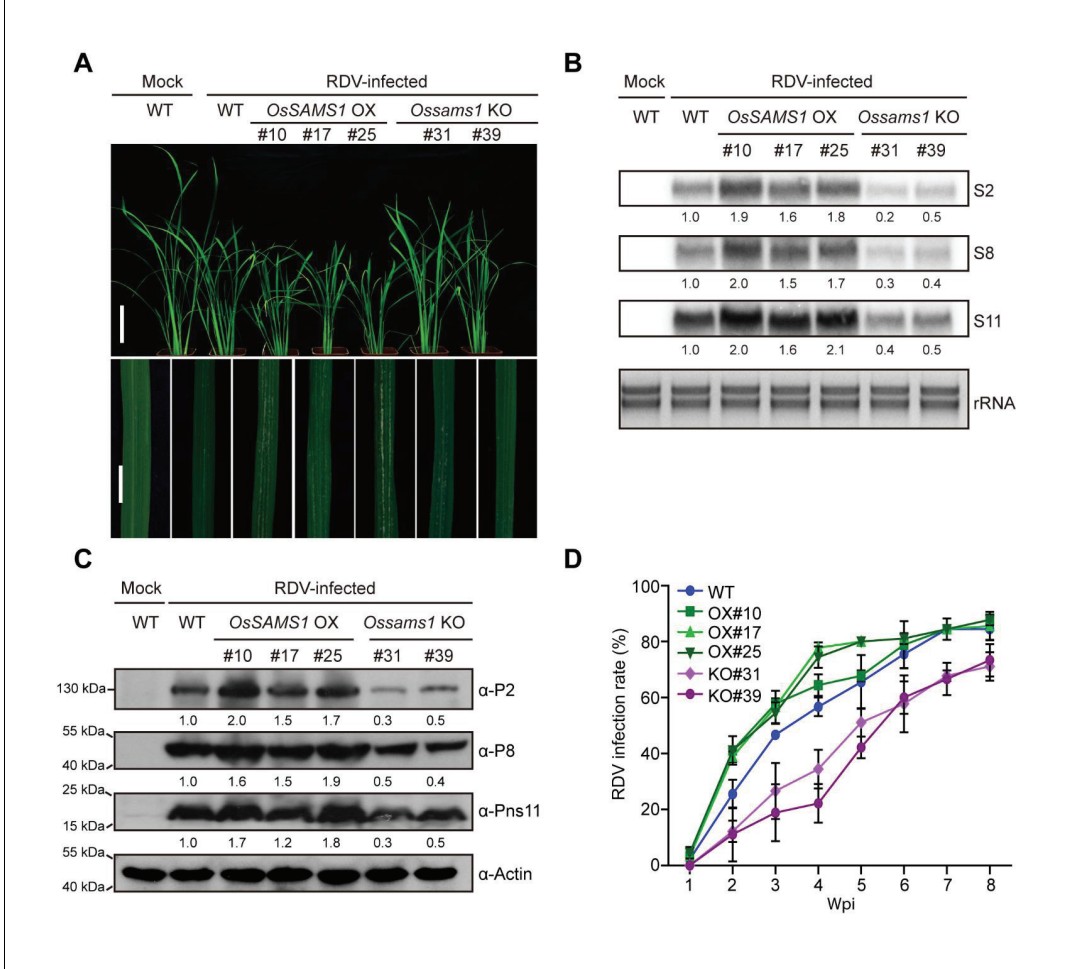

**Figure 4.** Overexpression of *OsSAMS1* enhances RDV infection whereas knockout of *OsSAMS1* reduces RDV infection.  (A) Symptoms of the mock-inoculated WT or RDV-infected WT (non-transgenic) plants and *OsSAMS1* OX/KO transgenic plants; images were taken at 4 wpi. Scale bars = 10 cm (upper panel) and 5 cm (lower panel). (B) Detection of RDV *S2*, *S8*, and *S11* genomic segments in the mock-inoculated WT or RDV-infected WT (non-transgenic) plants and in *OsSAMS1* OX/KO transgenic plants by northern blot. The blots were hybridized with radiolabeled riboprobes specific for each RNA segment. rRNAs were stained with ethidium bromide and served as loading controls. Tissues were collected at 4 wpi. (C) Detection of RDV P2, P8, and Pns11 proteins in the mock-inoculated WT or RDV-infected WT (non-transgenic) plants and in *OsSAMS1* OX/KO transgenic plants by western blot. Actin was probed and served as a loading control. Tissues were collected at 4 wpi. (D) The incidences of infection, which were determined by visual assessment of disease symptoms at 0–8 wpi of 30 individual plants for each case. Means and standard deviations were obtained from three independent experiments.

DOI: https://doi.org/10.7554/eLife.27529.016

The following figure supplement is available for figure 4:

**Figure supplement 1.** Overexpression of *OsSAMS1* enhances RDV infection while downregulation of *OsSAMS1* reduces RDV infection.
DOI: https://doi.org/10.7554/eLife.27529.017

parental *osein2* mutant, were less susceptible to RDV than was WT rice (*Figure 5—figure supplement 2*, *Supplementary file 1E*, *Supplementary file 2E*). Thus, we conclude that the ethylene-signaling pathway plays an important role in RDV infection and that blocking ethylene signaling would significantly enhance the antiviral defense response in rice.

## Ethylene is induced by viral infection and ethylene accumulation increases host susceptibility

We next investigated whether the interaction and activation between Pns11 and OsSAMS1 affects the levels of SAM, ACC, and ethylene in RDV-infected rice. We first performed a pull-down assay in WT and *Ossams1* KO lines, with and without RDV infection, using an anti-OsSAMS1 antibody

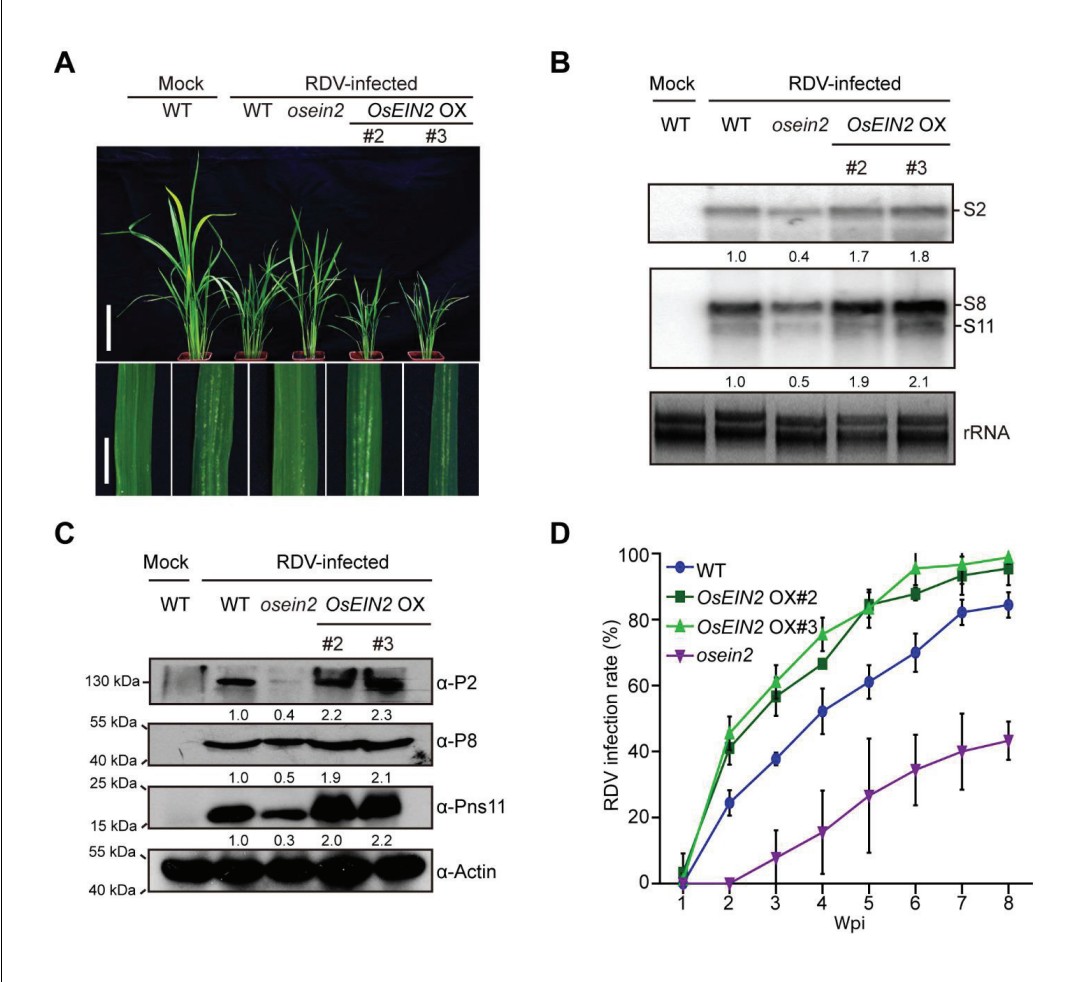

**Figure 5.** The ethylene-response mutant (*osein2*) shows increased tolerance of RDV infection whereas overexpression (OX) of *OsEIN2* results in enhanced susceptibility. (**A**) Symptoms of the mock-inoculated WT or RDV-infected WT (non-transgenic) plants, *osein2*, and *OsEIN2*-overexpression (OX) plants; images were taken at 4 wpi. Scale bars = 10 cm (upper panel) and 5 cm (lower panel). (**B**) Detection of RDV *S2*, *S8*, and *S11* genomic segments in the mock-inoculated WT or RDV-infected WT (non-transgenic) plants, *osein2*, and *OsEIN2* OX plants by northern blot. The blots were hybridized with radiolabeled riboprobes specific for each RNA segment. rRNAs were stained with ethidium bromide and served as loading controls. Tissues were collected at 4 wpi. (**C**) Detection of RDV P2, P8, and Pns11 proteins in the mock-inoculated WT or RDV-infected WT (non-transgenic) plants, *osein2*, and *OsEIN2* OX plants by western blot. Actin was probed and served as a loading control. Tissues were collected at 4 wpi. (**D**) The incidences of infection, which were determined by visual assessment of disease symptoms at 0–8 wpi of 30 individual plants for each case. Means and standard deviations were obtained from three independent experiments.

DOI: https://doi.org/10.7554/eLife.27529.018

The following figure supplements are available for figure 5:

**Figure supplement 1.** Phenotype and identification of *OsSAMS1* OE/*osein2* (J119) transgenic lines.
DOI: https://doi.org/10.7554/eLife.27529.019

**Figure supplement 2.** Overexpressed OsSAMS1 in *osein2* (J119) results in increased tolerance of RDV infection.
DOI: https://doi.org/10.7554/eLife.27529.020

**Figure supplement 3.** Detection of salicylic acid (SA)-, jasmonic acid (JA)- and ethylene-responsive genes.
DOI: https://doi.org/10.7554/eLife.27529.021

(*Figure 6A*). We found the loss of Pns11-OsSAMS1 interaction in *Ossams1* KO lines, with or without RDV infection. We then measured the SAM, ACC and ethylene levels in the same set of plants, and found that the RDV-induced increase of SAM, ACC and ethylene levels disappeared in *Ossams1* KO plants (*Figure 6B–D*). RNA-seq experiments on RDV-infected rice, *OsSAMS1* OX (OX#25) lines, *Ossams1* KO (KO#39) lines and *S11* OX (OX#11) lines were carried out and the differentially expressed genes in all comparable pairs were identified (*Supplementary file 3*). To determine

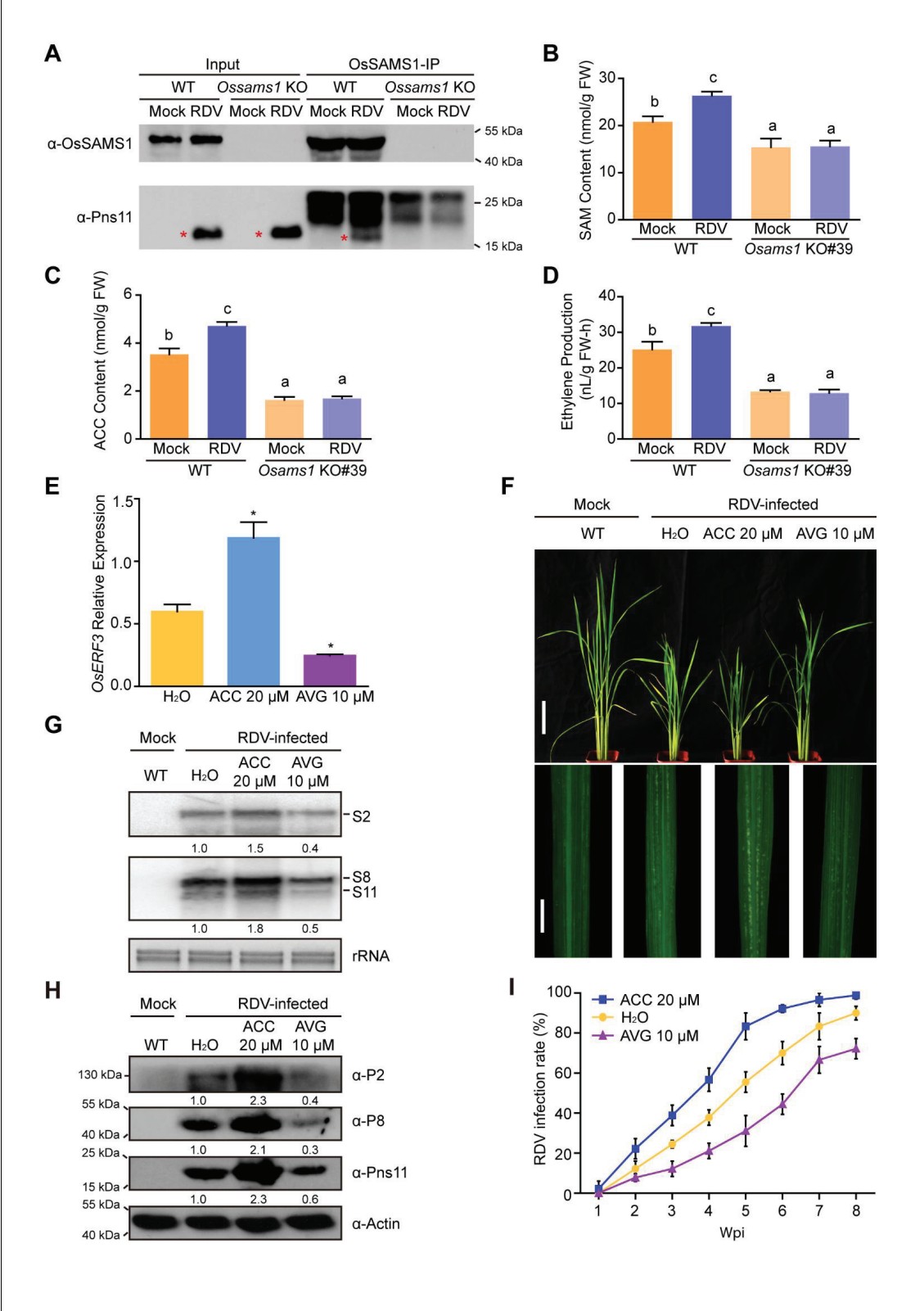

**Figure 6.** Ethylene is induced by RDV infection, which enhances the susceptibility of rice to RDV infection. (**A**) In *vivo* pull-down in WT and *Ossams1* KO lines, with or without RDV infection (40-d-old seedlings, 4 wpi), using anti-OsSAMS1 antibody. The red asterisk indicates the location of Pns11. (**B**) SAM content in WT and *Ossams1* KO lines, with or without RDV infection (40-d-old seedlings, 4 wpi). (**C**) ACC content in WT and *Ossams1* KO lines, with or without RDV infection (40-d-old seedlings, 4 wpi). (**D**) Ethylene content in WT and *Ossams1* KO lines, with or without RDV infection (40-d-old

*Figure 6 continued on next page*

*Figure 6 continued*

seedlings, 4 wpi). Tukey's honestly significant difference post hoc tests were performed for multiple comparisons. Letters indicate significantly different results, p<0.05. Data are from three replicates. FW, fresh weight. (**E**) Expression of *OsERF3* after 24 hr treatment with $H_2O$, 20 µM ACC, or 10 µM AVG. *OsERF3* was chosen as the positive control for ethylene responsiveness. The average (± standard deviation) values from three biological replicates are shown. Significant differences (*p<0.05) are based on Student's *t*-test. (**F**) Phenotypic comparison of mock-inoculated WT or RDV-infected rice plants pre-treated with $H_2O$, 20 µM ACC, or 10 µM AVG; images were taken at 4 wpi. Scale bars = 10 cm (upper panel) and 5 cm (lower panel). (**G**) Northern blot analysis of RDV *S2*, *S8*, and *S11* genomic segments. The blots were hybridized with radiolabeled riboprobes specific for each RNA segment. rRNAs were stained with ethidium bromide and served as loading controls. Tissues were collected at 4 wpi. (**H**) Western blot analysis of RDV P2, P8, and Pns11 proteins. Actin was probed and served as a loading control. Tissues were collected at 4 wpi. (**I**) The incidences of infection, which were determined by visual assessment of disease symptoms at 0–8 wpi of 30 individual plants for each case. Means and standard deviations were obtained from three independent experiments.

DOI: https://doi.org/10.7554/eLife.27529.022

The following figure supplement is available for figure 6:

**Figure supplement 1.** GO biological process over-representation in differentially expressed genes after RDV infection and in transgenic rice plants.

DOI: https://doi.org/10.7554/eLife.27529.023

whether the ethylene pathway was activated, Gene Ontology (GO) was used for analysis (*Figure 6—figure supplement 1*). Known ethylene-activated pathway genes were highly enriched in both RDV-infected and *OsSAMS1* OX transgenic rice, and depleted in *Ossams1* KO plants. These data indicate that RDV infection triggers ethylene synthesis and accumulation through the interaction of Pns11 and OsSAMS1, and the resultant activation of OsSAMS1.

To further elucidate the function of ethylene in RDV infection, 14-d-old seedlings were pretreated with 20 µM ACC, 10 µM AVG (aminoethoxyvinylglycine, an ethylene biosynthesis inhibitor) (*Chen et al., 2013a*), or $H_2O$ as a control for 1 day. We then sampled the treated plants and used qRT-PCR to analyze the expression of *OsERF3*, which could be a marker for the response to ethylene (*Qi et al., 2011*). *OsERF3* was significantly upregulated by ACC treatment and significantly downregulated by AVG treatment when compared to the $H_2O$-treated control (*Figure 6E*), demonstrating that the ACC and AVG treatments worked as expected. The treated plants were then inoculated with RDV-carrying or RDV-free leafhoppers (*Supplementary file 1F*). At 4 wpi, the ACC-treated plants showed greater susceptibility to RDV infection, displaying more severe disease symptoms and virus accumulation than the RDV-infected plants treated with $H_2O$. However, the AVG-treated plants exhibited enhanced disease tolerance, as shown by less virus accumulation and milder disease symptoms, when compared with the RDV-infected plants treated with $H_2O$ (*Figure 6F–H*). In addition, the rate of RDV infection in the ACC-treated plants increased much faster than that in the RDV-infected $H_2O$-treated plants, while the rate of infection in the AVG-treated plants increased more slowly in comparison with that in the $H_2O$-treated plants (*Figure 6I*, *Supplementary file 2F*). Taken together, these data suggest that RDV Pns11 interacts with OsSAMS1 and enhances its enzymatic activity, inducing ethylene biosynthesis and accumulation, which in turn enhances viral infection and host susceptibility.

## Discussion

Our results demonstrated that ethylene biosynthesis and signaling are critical for RDV infection and rice susceptibility. Overexpression of the RDV non-structural protein Pns11 increases rice susceptibility to viral infection (*Figure 1*). Furthermore, we found that Pns11 specifically interacts with OsSAMS1 and enhances its enzymatic activity in *vivo* and in *vitro* (*Figures 2* and *3*). Overexpression of *S11* or *OsSAMS1* increases the levels of SAM, ACC, and ethylene, whereas knockdown or knockout of *OsSAMS1* by RNAi or CRISPR/Cas9 reduces the level of SAM, ACC, and ethylene (*Figure 3*, *Figure 3—figure supplement 2*). Our results clearly indicated that an increase in ethylene production by overexpression of *OsSAMS1* decreases the host antiviral defense response and enhances RDV infection and accumulation in rice, whereas knockdown or knockout of *OsSAMS1* by RNAi or CRISPR/Cas9 reduces ethylene production, diminishes RDV accumulation, and increases the host antiviral defense response (*Figure 4*, *Figure 4—figure supplement 1*). In addition, plants that have compromised ethylene signaling are more tolerant to RDV infection (*Figure 5*, *Figure 5—figure supplement 2*). More importantly, RDV infection induces ethylene production, and the accumulation of

ethylene increases host susceptibility and enhances RDV infection and replication (*Figure 6*). Taken together, these results present a novel mechanism by which the virus highjacks host factors through enhancement of the enzymatic activity of SAMS1 and increasing ethylene production or signaling, thus reducing the host antiviral defense response and enhancing virus infection and accumulation (*Figure 7*). These findings provide a novel mechanism, deepen our understanding of the relationship between ethylene and viral infection, and will have a significant impact on our knowledge of the crosstalk between plant hormones and virus-host interactions.

We didn't find a strong difference between WT and *S11* OX#3 (*Figure 1*), especially in virus accumulation and infection rate. This is probably due to the relatively low expression level of Pns11 in this particular line (*Figure 1—figure supplement 1A,B*), which may be insufficient to induce a significant increase in ACC and ethylene production (*Figure 3B,C*). Furthermore, the hyper-susceptibility to RDV in Pns11-overexpressing plants was more prominent prior to 3 wpi (*Supplementary file 2A*). This is easily explained by the fact that enhanced susceptibility allowed more Pns11 transgenic plants to show more conspicuous symptoms at earlier time points. This does not, however, prevent WT plants from becoming symptomatic at later time points, thus catching up with the transgenic plants in the proportion of plants that are infected.

RDV infection affects a number of genes that are interact with the signaling pathways of plant hormones such as JA, ethylene, gibberellin, and auxin. A mutation of a NAC-domain transcription factor, which regulates JA signaling, confers strong tolerance to RDV infection in rice (*Yoshii et al., 2009 , 2010*). Previous studies in our lab have indicated that RDV-encoded P2 interacts with *β-ent*-kaureen oxidases to reduce gibberellic acid synthesis, resulting in dwarfism (*Zhu et al., 2005*). P2 also reprograms the initiation of auxin signaling through interaction with OsIAA10, thus enhancing viral infection and pathogenesis (*Jin et al., 2016*). Here, we report a mechanism by which RDV-encoded Pns11 promotes ethylene production to enhance plant susceptibility to viral infection (*Figure 7*). RNA-seq of RDV-infected rice also showed a regulation of hormone-responsive genes (*Figure 6—figure supplement 1*). Thus, it appears that RDV may interfere with phytohormone pathways to counteract plant immune responses. This complex crosstalk and these hormonal changes may be regulated by RDV infection, especially through interactions with host factors.

Ethylene regulates numerous developmental processes and adaptive stress responses in plants (*van Loon et al., 2006*; *Broekgaarden et al., 2015*; *Kazan, 2015*). During biotic stress, ethylene is mainly responsible for defense against necrotrophic pathogens (*Pieterse et al., 2012*; *Broekgaarden et al., 2015*) and plays a dual role in the plant defense signaling pathway. In some cases, ethylene is used by pathogens as a virulence factor to enhance pathogenesis, whereas in other cases, ethylene aids in the alleviation of stress. Generally, the plant defense responses

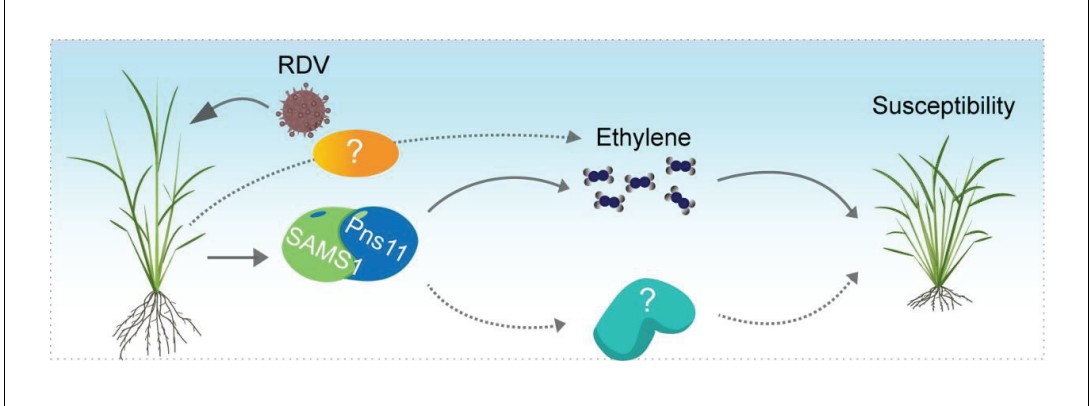

**Figure 7.** Possible model for Pns11 enhancement of OsSAMS1 enzymatic activity to increase ethylene levels and enhance host susceptibility to viral infection. The model proposes that the RDV-encoded protein Pns11 specifically interacts with OsSAMS1 to enhance its enzymatic activity, resulting in a corresponding increase in ethylene, and thus enhancing rice susceptibility to RDV infection. However, RDV infection may affect other pathways and SAM is also the precursor of polyamine and a methyl donor for methylation, therefore, other pathways may be involved in RDV pathogenesis and need further study.

DOI: https://doi.org/10.7554/eLife.27529.024

regulated by ethylene depend on the specific host-pathogen interaction and the crosstalk between multiple signals (*Broekaert et al., 2006*; *van Loon et al., 2006*; *Denancé et al., 2013*; *Alazem and Lin, 2015*; *Wang, 2015*). Although the function of ethylene has been addressed in various host-pathogen interactions (*Iwai et al., 2006*; *Shen et al., 2011*; *Groen et al., 2013*; *Helliwell et al., 2013*; *Kim et al., 2013*; *Yang et al., 2017*), the role of ethylene and its underlying mechanisms in plant-virus interactions are not well understood, with only a few reports on the involvement of ethylene and ethylene signaling in virus-host interactions (*Marco and Levy, 1979*; *Knoester et al., 2001*; *Huang et al., 2005*; *van Loon et al., 2006*; *Love et al., 2007*; *Endres et al., 2010*; *Haikonen et al., 2013*). In order to gain a deeper insight into the mechanism, we analyzed some defense or hormone-responsive genes known to function in SA-, JA- or ethylene-associated pathways and found pathogenesis-related protein 1b (*OsPR1b*) to be highly expressed in *osein2, OsSAMS1* RNAi and KO lines but expressed at reduced levels in *OsEIN2* OX, *OsSAMS1* OE lines and *S11* OX lines (*Figure 5—figure supplement 3*) (*Shen et al., 2011*). *PRs* are known to be induced by pathogen infection and involved in responses to many plant phytohormones, disease resistance and general adaptation to stressful environments (*Huang et al., 2005*; *Alazem and Lin, 2015*). Analysis of RDV microarray data and our RNA-seq data revealed that *OsPR1b* was also induced after virus infection (*Satoh et al., 2011*) (*Supplementary file 3*), indicating a role for *OsPR1b* of rice in defense against RDV. Further studies will improve our understanding of the function of ethylene in plant defense responses and its underlying mechanisms.

SAMS is a key enzyme in plants and catalyzes the conversion of ATP and L-Met into SAM (*Roje, 2006*). Expression of the *SAMS* gene is induced by a number of biotic and abiotic stresses and confers increased tolerance to various stresses (*Kawalleck et al., 1992*; *Boerjan et al., 1994*; *Gómez-Gómez and Carrasco, 1998*). Overexpression of *SAMS* genes in plants alters development (*Boerjan et al., 1994*) and confers increased tolerance to abiotic stress. Knockdown of *SAMS* genes affects plant development (*Boerjan et al., 1994*), leading to late flowering and abnormal methylation in rice (*Li et al., 2011*), and is also related to viral RNA stabilization and accumulation in *N. benthamiana* (*Ivanov et al., 2016*). We found that knockdown or knockout of *OsSAMS1* resulted in abnormal phenotypes (*Figure 3—figure supplement 1*). These studies indicate that SAMS is a broad-spectrum signaling molecule that regulates plant responses to various stresses. SAM acts as the precursor in the biosynthesis of polyamines (PAs) and ethylene (*Roje, 2006*). The involvement of PAs and their metabolism in defense responses against diverse viruses has also been demonstrated (*Yoda et al., 2003*; *Mitsuya et al., 2009*). We provide strong evidence that RDV infection activates OsSAMS1 and increases the production of SAM and ethylene (*Figures 2*, *3* and *6*), and that disruption of the ethylene signaling pathway enhances rice tolerance to RDV infection (*Figure 5*, *Figure 5—figure supplement 2*). GO analysis of our RNA-seq data also showed that the class of ethylene-activated pathway genes were highly enriched in both RDV-infected and *OsSAMS1* OX transgenic rice, and that knockout of *Ossams1* significantly affected the ethylene biosynthetic process. Interestingly, although genes in the hormone-mediated signaling pathway category were enriched in *S11* OX transgenic lines, those in the ethylene-activated pathway were not. This is consistent with the observation that Pns11 overexpression enhances OsSAMS1 activity without upregulating its mRNA. In addition, relative to the Pns11 expression level in RDV infection, the transgenically expressed Pns11 level in *S11* OX lines was probably low and insufficient to induce significant changes in ethylene pathway genes. Results from early microarray analyses and our RNA-seq data (*Figure 6—figure supplement 1*, *Supplementary file 4*) (*Satoh et al., 2011*; *Do et al., 2013*) showed no obvious changes in the polyamine pathway in RDV-infected rice compared to the control. These results indicated that the ethylene pathway regulated by Pns11 and OsSAMS1 interaction may be the major determinant of RDV pathogenesis, but an additional mechanism involving other RDV proteins cannot be ruled out at this point (*Figure 7*).

Here, we report that Pns11 enhances OsSAMS1 activity in *vitro* and in *vivo*, but the underlying mechanism remains unknown. A previous study suggested that enhanced substrate affinity may increase enzyme activity (*Toroser et al., 1999*); but we found that Pns11 did not alter the affinity of OsSAMS1 to the substrate L-Met or ATP (*Figure 2—figure supplement 5*). Crystal structures of SAMS have been elucidated from other non-plant organisms, and SAMS isoenzymes appear as homotetramers, dimers, or heterooligomers (*Markham and Pajares, 2009*). Although the crystal structure of plant SAMS has not been reported to date, the high sequence similarity of plant SAMS to the known SAMS sequences (*Figure 2—figure supplement 6*) (*Li et al., 2013*) suggests that

OsSAMS1 may function as dimers or tetramers. We found that OsSAMS1 exists in a high molecular weight form in RDV-infected rice (*Figure 2—figure supplement 7*), and we propose that the interaction of Pns11 with OsSAMS1 may promote oligomerization of OsSAMS1 to the most active form, which may increase its enzymatic activity and lead to the production of more SAM. The mechanism through which Pns11 activates OsSAMS1 remains unknown and requires further exploration.

## Materials and methods

### Plant growth and virus inoculation

Plant growth and virus inoculation were carried out as previously described (*Wu et al., 2015*; *Jin et al., 2016*). Rice seedlings were grown in a greenhouse at 28–30°C for 2 weeks and plants at the third-leaf stage were inoculated with 2–3 viruliferous leafhoppers per plant for 2 days. The insects were then removed and the rice seedlings were maintained under the same growing conditions. At 4 weeks post inoculation, when the viral symptoms appeared on the new leaves, the seedlings were photographed and harvested. A minimum of 30 rice seedlings were used for each sample. The index of non-preference for each line was characterized by the mean number of settled insects on each seedling (*Supplementary file 1*) as previously described (*Jin et al., 2016*). The number of plants with symptoms for each line was recorded every week and statistical analysis of the infection rates was carried out (*Supplementary file 2*).

### Vector construction and rice transformation

The entire open reading frames (ORFs) of *OsSAMS1* and *S11* were amplified by RT-PCR and then introduced into the *pCam2300:Ubi:OCS* vector to yield *pCam2300:Ubi:Flag OsSAMS1* and *pCam2300:Ubi:HA S11*. The *pUCC-OsSAMS1* was used to create the *OsSAMS1* RNAi knockdown transgenic lines. The *OsSAMS1* knockout construct was constructed as previously described (*Miao et al., 2013*). The resulting constructs were used for transformation via *Agrobacterium* (Bio-Run, Wuhan, China). All primers used in this assay are listed in *Supplementary file 5A and B*.

### Quantification of SAM, ACC, and ethylene from rice leaves

Leaves of the same position were cut into 8 cm pieces and six pieces were placed into a 50 mL glass vial with distilled water sealed with a gas-proof septum. After imbibition in a growth cabinet at 28°C for 48 hr, a 0.1 mL gas sample was withdrawn from the head space of each bottle using a gas-tight syringe (Hamilton), and the ethylene concentration was determination by gas chromatography (Agilent 6890N) equipped with an activated alumina column and flame ionization detectors. A six-point standard ethylene curve with concentration ranging from 0.5 to 3.0 $\mu L \cdot L^{-1}$ was used for the calibration. The quantified data, divided by fresh weight and time, were converted to specific activities.

ACC was extracted from the same leaf tissues used for quantifying ethylene contents and ground in liquid nitrogen using a mortar and pestle, then stirred with 80% (v/v) ethanol ($2 \text{ mL} \cdot g^{-1}$ fresh weight) and the supernatant was evaporated to dryness. The residue was then resuspended in water. The ACC concentration in the supernatant was determined directly by chemical conversion to ethylene as described previously (*Lizada and Yang, 1979*; *Chen et al., 2013a*).

SAM was extracted from rice leaves with 5% (w/v) trichloroacetic acid (TCA, Sigma-Aldrich). For each extraction, frozen tissue powder (0.2 g) was homogenized with extraction solution (1 mL) for 15 min at 4°C and the homogenate was centrifuged at 10,000x *g* for 15 min followed by another centrifugation at 13,000x *g* for 15 min. The supernatant was collected by filtration through a 0.45 µm pore-size Millipore filter. All steps were carried out on ice or at 4°C (during centrifugation) to prevent SAM degradation (*Van de Poel et al., 2010*). 5 µl of supernatant was used for analysis of SAM using LC-MS/MS (Agilent UPLC 1290 MS/MS 6495) and the conditions are listed in *Supplementary file 5C*. Five standard concentrations of S-adenosyl-L-methionine solutions (0.000625, 0.00125, 0.0025, 0.005, and 0.01 $mg \cdot mL^{-1}$) were prepared for the standard curve.

### Yeast two-hybrid assay

The DUALhunter starter kit (Dualsyetems Biotech) was used for the yeast two-hybrid assays. All protocols were carried out according to the manufacturer's manual. The rice cDNA library was constructed in prey plasmid *pPR3-N* using an EasyClone cDNA library construction kit (Dualsystems

Biotech), and the bait plasmid was constructed by inserting full-length RDV-encoded Pns11 into the *pDHB1* vector. After library screening, positive clones were selected on SD quadruple dropout (QDO) medium (SD/-Ade/-His/-Leu/-Trp) and prey plasmids were isolated from these clones for sequencing. To further distinguish positive from false-positive interactions and to confirm the interaction of bait and prey proteins, we co-transformed the two plasmids into yeast strain NMY51 and detected β-galactosidase activity with an HTX Kit (Dualsystems Biotech).

## Co-IP assay

The ORF PCR products of *S11* and *OsSAMS1/2/3* were inserted into the *pCam2300:35S:OCS* vector (*Wu et al., 2015*) to yield *pCam2300:35S:HA S11* and *pCam2300:35S:Flag OsSAMS1/2/3*. The constructs were then co-infiltrated into *N. benthamiana* leaves by agroinfiltration. Leaves were harvested 3 days post-infiltration and total proteins were extracted with co-IP buffer (50 mM Tris-Cl pH 7.5, 150 mM NaCl, 10% glycerol, 0.5 mM EDTA, 0.5% NP-40, and $1 \times$ protease inhibitor cocktail). After incubation on ice for 30 min, plant extracts were sonicated and then centrifuged. Cleared extract was combined with anti-Flag or anti-HA antibodies together with recombinant protein G-Sepharose 4B (Invitrogen) and incubated for 3 hr at 4°C. After washing five times with co-IP buffer, agarose beads were collected by centrifugation (2000x *g* for 2 min) and then resuspended in protein extraction buffer. Proteins were separated by SDS-PAGE and detected with the corresponding antibody.

## Firefly luciferase complementation imaging assay

The ORFs of *S11* and *OsSAMS1* were inserted into the *pCAMBIA1300-nLUC* and *pCAMBIA1300-cLUC* vectors, respectively (*Jin et al., 2016*). The constructs were then transformed into *Agrobacterium tumefaciens* strain EHA105 and cultured to $OD_{600} = 0.5$, combined with equal volumes of the adjusted culture for specific groups as shown in the figure legends, and incubated at room temperature without shaking for 3 hr followed by infiltrating into *N. benthamiana* leaves. The LB 985 Night-SHADE system (Berthold Technologies) was used for luciferase activity detection 3 days after infiltration.

## Gel filtration assay

The samples were prepared as described in the previous sections using 2 g of rice leaves and 3 mL of co-IP buffer. The lysates were then filtered through a 0.22 μm filter. 750 μl of total protein was loaded onto a Superdex 200 10/300 GL column (GE Healthcare) and 250 μl fractions were collected at 0.3 ml·min$^{-1}$.

## Bimolecular fluorescence complementation assay

Bimolecular fluorescence complementation (BiFC) was carried out using previously described vectors and methods (*Yang et al., 2011*). The ORFs of *S11* and *OsSAMS1* were inserted into the BiFC expression vectors p2YN and p2YC, respectively. The constructs were mixed 1:1 immediately prior to co-infiltrate into *N. benthamiana* leaves by agroinfiltration. Leaf tissue was analyzed 3 days post-inoculation by microscopy using a Zeiss LSM710 confocal laser scanning microscope equipped with a C-Apochromat 40X/1.20NA water immersion objective. Images were photographed under either white light or UV light and a Chroma filter with a 450- to 490 nm excitation wavelength and 515 nm emission wavelength was used to record YFP. All primers used in this assay are listed in *Supplementary file 5A and B*.

## In *vivo* pull-down assay

Samples were extracted with IP buffer (50 mM Tris-Cl pH 7.5, 150 mM NaCl, 10% glycerol, 0.5 mM EDTA, 0.5% NP-40, and $1 \times$ protease inhibitor cocktail). After incubation on ice for 30 min, the pull-down assay was performed utilizing a Beaver Beads Protein A/G Matrix Immunoprecipitation kit (Beaver Nano-Technologies Co. China) following the manufacturer's instructions with anti-OsSAMS1 antibody (Abgent, Suzhou, China). Proteins were separated by SDS-PAGE and detected with the corresponding antibody.

## Protein expression, purification, and enzyme activity assays in *vitro*

*OsSAMS1* and *OsSAMS2* were amplified by PCR and then inserted into the pGEX vector (GE Healthcare Life Sciences) and expressed as glutathione S-transferase fusion proteins (GST-OsSAMS1/ OsSAMS2) in *E. coli* BL21 cells. After purification by glutathione-agarose chromatography, the GST tag was removed. *S11*, *S9*, and *GFP* were also amplified by PCR and introduced into the pMal-p2x vector, which fused a maltose-binding protein at the N-terminal. The constructs were transformed into *E. coli* BL21 cells and purified by amylose affinity chromatography. Primers used for amplification of *OsSAMS1*, *OsSAMS2*, *S11*, *S9*, and *GFP* prior to insertion in expression vectors are listed in *Supplementary file 5A and B*.

Indirect assays of OsSAMS1 and OsSAMS2 activity were carried out according to the scheme presented in *Figure 2E* using SAM production by OsSAMS1/2 as a measure of OsSAMS1/2-catalyzed L-Met with ATP yielding SAM. Mixtures containing 30 pmol OsSAMS1/2 and various amounts of Pns11 (P9 or GFP) in a total volume of 15 µl were pre-incubated at 30°C for 20 min. Mixtures were then added to reactions containing final concentrations of 100 mM Tris-HCl pH 8.0, 200 mM KCl, 10 mM $MgCl_2$, 1 mM DTT, 3.3 mM ATP, and 5 µCi L-[$^{35}$S]-methionine (1175 Ci/mmol). The reactions were incubated at 30°C for another 20 min, then OsSAMS1/2 activity was terminated by addition of 1.5 µl of 0.5M EDTA. SAM production was analyzed by thin layer chromatography on polyethyleneimine cellulose HPTLC plates developed with n-butyl alcohol:acetic acid:water (12:3:5, v/v) (*Kim et al., 2003*). After chromatography, radioactive signals on plates were quantitated using a phosphorimager (Typhoon FLA900, GE Healthcare).

## Northern blot and quantitative RT-PCR analysis

For northern blot, 15 µg of total RNA was extracted from rice plants with Trizol (Invitrogen), separated by 1% formaldehyde agarose gel and transferred to Hybond-N +membranes that were then cross-linked and dried as previously described (*Wu et al., 2015*). The 500 bp probes that were partially complementary to RDV-encoded *S2*, *S8*, and *S11* were labeled with α-$^{32}$P-dCTP. The sequence of the probes and the primers are listed in *Supplementary file 5D*. For RT-PCR, total RNA (2 µg) was reverse transcribed into cDNA by SuperScript III Reverse Transcriptase (Invitrogen). qRT-PCR amplification was performed in 20 µL reactions containing 4 µL of 20-fold diluted cDNA, 0.5 µM of each primer, and 10 µL of SYBR Green PCR Master Mix (Toyobo). The expression was normalized to that of *EF-1α*. Primer sequences in this assay are listed in *Supplementary file 5D*.

## RNA-seq analysis

Total RNAs were extracted from RDV-infected rice plants (4 wpi, 42-d-old seedlings) and transgenic rice lines (42-d-old seedlings) using the RNeasy plant mini kit (Qiagen). The RNA-seq analyses were performed at Bionova Company. Libraries were constructed through adaptor ligation and were subjected to pair-ended sequencing with a 150-necleotide reading length. FastQC software was used to access the quality of raw sequencing reads. After removing adaptor and low-quality reads, clean reads were mapped to rice genome MSU7.0 using TopHat. Responsive genes were identified by reads per kilobase per million reads (RPKM) and edgeR software was used to identify differential expressed genes. The multiple-testing adjusted P-value (FDR < 0.05) and fold change (FC >2) was used to determine whether the gene was significantly differentially expressed or not. Three biological replicates were used, and their repeatability and correlation were evaluated by the Pearson's Correlation Coefficient (*Schulze et al., 2012*). The outputs of RNA-seq analysis used in this study (series number GSE102366) are available at NCBI-GEO.

## MST assays

The MST assay was performed as previously described (*Jin et al., 2016*). OsSAMS1 protein was labeled with the red fluorescent dye NHS according to the Monolith NT Protein Labeling Kit RED-NHS instructions (NanoTemper Technologies GmbH; München, Germany). The concentration of NHS-labeled OsSAMS1 was held constant at 1.25 µM, whereas the concentrations of L-Met and ATP were gradient-diluted (L-Met: 200,000 nM, 100,000 nM, 50,000 nM, until it reached 97.7 nM; ATP: 50,000 nM, 25,000 nM, 12,500 nM, until it reached 24.4 nM). After a short incubation, the samples were loaded into MST standard treated glass capillaries. Measurements were performed at 25°C in buffer containing 20 mM Tris pH 8.0 and 150 mM NaCl using 40% LED power and 20% MST power.

The assays were repeated three times for each affinity measurement. Data analyses were performed using the Nanotemper Analysis and OriginPro 8.0 software provided by the manufacturer. In affinity affecting assays, 2.5 μM of MBP-Pns11 or MBP was added to 1.25 μM NHS-labeled OsSAMS1 and gradient-diluted concentrations of L-Met or ATP. After a short incubation, the samples were loaded into MST standard treated glass capillaries for MST analysis as described above.

## Acknowledgements

We thank Drs Lijia Qu, Hongwei Guo, Rongfeng Huang and Feng Qu for critical comments, Yijun Zhou (Jiangsu Academy of Agricultural Sciences, Nanjing, China) and his team for providing us with isolated fields for rice planting, Kang Chong (Institute of Botany, Chinese Academy of Sciences, Beijing, China) for providing materials, and the Core Facilities at College of Life Sciences, Peking University for assistance with Ultra Performance Liquid Chromatography-Mass spectrometry (UPLC-MS/MS) work. We are grateful to Dr. Gaojun Liu for her help with data acquisition and analysis.

## Additional information

### Funding

| Funder | Grant reference number | Author |
|---|---|---|
| National Natural Science Foundation of China | 31530062 | Yi Li |
| National Natural Science Foundation of China | 31420103904 | Yi Li |
| Ministry of Science and Technology of the People's Republic of China | National Basic Research Program 973, 2014CB138400 | Yi Li |
| Ministry of Agriculture of the People's Republic of China | Transgenic Research Program 2016ZX08010-001 | Yi Li |

The funders had no role in study design, data collection and interpretation, or the decision to submit the work for publication.

### Author contributions

Shanshan Zhao, Conceptualization, Data curation, Formal analysis, Funding acquisition, Investigation, Visualization, Writing—original draft, Writing—review and editing; Wei Hong, Conceptualization, Data curation, Formal analysis, Investigation, Visualization, Methodology, Writing—original draft, Writing—review and editing; Jianguo Wu, Yu Wang, Formal analysis, Investigation, Visualization, Methodology; Shaoyi Ji, Shuyi Zhu, Chunhong Wei, Formal analysis, Methodology; Jinsong Zhang, Yi Li, Resources, Formal analysis

### Author ORCIDs

Jianguo Wu (iD) https://orcid.org/0000-0001-5025-5026
Yi Li (iD) http://orcid.org/0000-0002-0258-3452

### Decision letter and Author response

Decision letter https://doi.org/10.7554/eLife.27529.035
Author response https://doi.org/10.7554/eLife.27529.036

## Additional files

### Supplementary files

• Supplementary file 1. Non-preference test.
DOI: https://doi.org/10.7554/eLife.27529.025

• Supplementary file 2. Record of the number of rice plants showing RDV symptoms at time course and infection rates statistical analysis.

DOI: https://doi.org/10.7554/eLife.27529.026

• Supplementary file 3. Differentially expressed genes of RDV-infected and *OsSAMS1* and *S11* transgenic rice RNA-seq analysis.
DOI: https://doi.org/10.7554/eLife.27529.027

• Supplementary file 4. Responses of genes related to polyamine by RDV infection.
DOI: https://doi.org/10.7554/eLife.27529.028

• Supplementary file 5. (A) Constructs in this study. (B) Primers for plasmids constructions in this study. (C) LC-MS/MS conditions. (D) Primers for RNA gel blot probes, real-time PCR, and semi-quantitative RT-PCR.
DOI: https://doi.org/10.7554/eLife.27529.029

• Transparent reporting form
DOI: https://doi.org/10.7554/eLife.27529.030

### Major datasets

The following dataset was generated:

| Author(s) | Year | Dataset title | Dataset URL | Database, license, and accessibility information |
|---|---|---|---|---|
| Zhao S, Hong W, Wu J, Wang Y, Ji S, Zhu S, Wei C, Zhang J, Li Y | 2017 | A viral protein promotes host SAMS1 activity and ethylene production for the benefit of virus infection | https://www.ncbi.nlm.nih.gov/geo/query/acc.cgi?acc=GSE102366 | Publicly available at the NCBI Gene Expression Omnibus (accession no. GSE102366) |

The following previously published dataset was used:

| Author(s) | Year | Dataset title | Dataset URL | Database, license, and accessibility information |
|---|---|---|---|---|
| Shimizu T, Sasaya T, Kondoh H, Satoh H, Kimura S, Omura T, Kikuchi S | 2011 | Transcriptome analysis of rice infected with three RDV strains | https://www.ncbi.nlm.nih.gov/geo/query/acc.cgi?acc=GSE24937 | Publicly available at the NCBI Gene Expression Omnibus (accession no. GSE24937) |

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
