## [Decision Letter]

Thank you for submitting your article "A viral protein promotes host SAMS1 activity and ethylene production for the benefit of virus infection" for consideration by *eLife*. Your article has been reviewed by three peer reviewers, and the evaluation has been overseen by a Reviewing Editor and Detlef Weigel as the Senior Editor. The reviewers have opted to remain anonymous.

The reviewers have discussed the reviews with one another and the Reviewing Editor has drafted this decision to help you prepare a revised submission.

Summary:

In this work, the authors studied the pathogenesis of rice dwarf virus (RDV), which infects the major crop rice through the green rice leafhopper and causes significant economic losses every year. The authors focused on the functional characterization of the RDV non-structural protein Pns11, a protein that is found in the viroplasm, and has previously been reported to bind to nucleic acids in a Zn-dependent and sequence-unspecific manner. Pns11 has been implicated in the RDV virulence.

To study the function of Pns11, the authors generated transgenic rice plants that express Pns11 and performed RDV infection assays to assess the role of Pns11 in pathogenesis. For this, the authors used the green leafhopper inoculation system, which reproduces the natural infection conditions. Their analysis lead them to propose that ethylene functions as a susceptibility factor for RDV pathogenesis in rice. They identified OsSAMS1 as a Pns11-interacting protein, which could mean that the ethylene pathway is involved (something that is supported by the use of an ethylene inhibitor and an EIN2 overexpressor).

Essential Revisions:

The reviewers were concerned that the authors did not consider alternative explanations for their data. For example, the SAM cycle affects multiple pathways; the ethylene pathway may not be specifically involved here or there may be an effect of the synthesis of polyamines (in addition to the effect on ethylene). It’s important not to overinterpret or bias the explanations of the data. A number of questions arose that require experiments to answer.

1) What is the mechanism of a potential effect on ethylene production?

Virus infection appears to result in the increase in SAM, ACC, and ethylene (Figure 6). However, whether this is due to the interaction of PNS11 with the SAM synthase OsSAMS1 has not been addressed.

To show the causal relationship the authors should use their Ossams1 KO lines (Figure 4), and show that there is no longer an effect on the ethylene pathway when infection occurs.

2) Is there an interaction between Pns11 and osSAMS1 during RDV infection? This needs to be addressed using a time course. The necessary antibody reagents seem to be available for performing these experiments.

3) Are there gene expression changes that reflect activation of the ethylene pathway? This could be addressed using RNA-seq data from osSAMS1 and Pns11 transgenic plants. Changes in gene expression in the ethylene pathway may show a similar pattern as that seen during the virus infection. If not, the model should be revised.

4) The authors repeatedly cite Satoh et al., (2011) as showing that ethylene related genes are induced following RDV infection (Introduction, Abstract). But that cited paper actually says the opposite: "However, the genes for ET and SA synthesis were not strongly activated by RDV infection." The authors finding that SAMS1 is regulated post-transcriptionally is consistent with the lack of transcriptional changes noted by Satoh et al.,. The rationale for stating that Satoh et al., demonstrate the induction of ethylene related genes needs to be justified (GO analysis of microarray data?) or else interpretation of these earlier results revised and discussed accordingly.

5) The authors cannot state that virus accumulation 'significantly' increased (subsection “RDV infection”) as they do not provide statistics. Also, they cannot state "the infection rates were much higher in the S11 OX lines compared to Wt (Figure 1)" as the difference is actually rather small, and line OX#3 is the same as Wt.

6) Supplementary file 2 data suggests that the effect is observed before 2-3 wpi; but at later time points there is no difference between wild type and OX plants. Authors should comment on this; because it is not providing complete resistance to RDV, it is providing tolerance to the RDV suggesting ethylene is not the only factor required for RDV pathogenesis.

7) Figure legends – lack details on timing of tissue collection and days after infection, etc.

8) Text in terms resistance should be toned down; it is really tolerance, not resistant. "Inhibits RDV infection" – is misleading because it reduces infection compared to the wild type plants. subsection “Ethylene is induced by viral infection and ethylene accumulation increases host susceptibility”: "RDV infection triggers ethylene synthesis and accumulation through the interaction of Pns11 and OsSAMS1, and resultant activation of OsSAMS1". Although there is indirect data for this conclusion in the paper, there is no direct evidence that this happens during RDV infection.

[Editors' note: further revisions were requested prior to acceptance, as described below.]

Thank you for resubmitting your work entitled "A viral protein promotes host SAMS1 activity and ethylene production for the benefit of virus infection" for further consideration at *eLife*. Your revised article has been favorably evaluated by Detlef Weigel as the Senior editor, a Reviewing editor, and two reviewers.

The manuscript has been improved but there are some remaining issues that need to be addressed before acceptance, as outlined below:

You should revise the text (subsection “OsSAMS1 enzymatic activity”). It is misleading to indicate that proteins present in the same fraction means they are interacting. Your BiFC data and Figure 6—figure supplement 1 does show complex formation. Please be sure to carefully revise the text and be sure to note what each experiment really shows.

Reviewer #1:

The authors have responded appropriately to the concerns raised by the reviewers in their revised manuscript, adding additional supporting data and making appropriate textual changes.

Reviewer #2:

Authors have addressed most of the issues raised in the last reviews. However, authors fail to respond to this comment:

– Figure 2 – This reviewer failed to understand how the authors conclude interaction based on fractionation? This should be a simple IP experiment considering the authors have antibodies to Pns11 and OsSASM1. IP with OsSAMS1 antibody and probe for Pns11, in mock and RDV infected tissues. Alternatively, IP with Pns11 antibody and probe for OsSAMS1.

---

## [Author Response]

1) What is the mechanism of a potential effect on ethylene production?Virus infection appears to result in the increase in SAM, ACC, and ethylene (Figure 6). However, whether this is due to the interaction of PNS11 with the SAM synthase OsSAMS1 has not been addressed.To show the causal relationship the authors should use their Ossams1 KO lines (Figure 4), and show that there is no longer an effect on the ethylene pathway when infection occurs.

We thank this reviewer for the insightful comments. To address this concern, we first carried out experiments to confirm the loss of Pns11-OsSAMS1 interaction in Ossams1 KO lines, with or without RDV infection. This is done through a pull-down assay, the results of which are described in subsection “Ethylene is induced by viral infection andethylene accumulation increases host susceptibility” of the revised manuscript, and shown in Figure 6—figure supplement 2A. Specifically, mock-inoculated WT and Ossams1 KO plants, as well as RDV-infected WT and Ossams1 KO plants, were subjected to protein extraction, followed by immune pull-down with an anti-OsSAMS1 antibody. Pns11 was pulled down from RDV-infected WT extracts, but not Ossams1 KO extracts.

We then measured the SAM, ACC and ethylene levels in the same set of plants, and found that the RDV-induced increase of SAM, ACC and ethylene levels disappeared in Ossams1 KO plants (Figure 6—figure supplement 2B-D). These results are exactly as predicted by the reviewer, thus addresses his/her concern.

2) Is there an interaction between Pns11 and osSAMS1 during RDV infection? This needs to be addressed using a time course. The necessary antibody reagents seem to be available for performing these experiments.

This concern is addressed in part by the pull-down assay described above. We further bolstered this conclusion by performing the time course pull-down experiments recommended by this reviewer, which showed a consistent OsSAMS1-Pns11 interaction starting three weeks post inoculation (subsection “Ethylene is induced by viral infection andethylene accumulation increases host susceptibility” in the revised manuscript and Figure 6—figure supplement 1).

3) Are there gene expression changes that reflect activation of the ethylene pathway? This could be addressed using RNA-seq data from osSAMS1 and Pns11 transgenic plants. Changes in gene expression in the ethylene pathway may show a similar pattern as that seen during the virus infection. If not, the model should be revised.

RNA-seq experiments have now been carried out and the results of which incorporated in the revised manuscript (See subsection “Ethylene is induced by viral infection andethylene accumulation increases host susceptibility” in the revised manuscript). Specifically, the gene expression profiles of RDV-infected rice, OsSAMS1 OX lines, Ossams1 KO lines and S11 OX lines were analyzed to identify differentially expressed genes in all comparable pairs (See Supplementary file 3). To determine whether the ethylene pathway was activated by RDV infection, Gene Ontology (GO) was used for analysis (See Figure 6—figure supplement 3). Known ethylene-activated pathway genes were highly enriched in both RDV-infected and OsSAMS1 OX transgenic rice, and depleted in Ossams1 KO plants. Interestingly, although genes in the hormone mediated signaling pathway category were enriched in S11 OX transgenic lines, those in the ethylene-activated pathway were not. This is consistent with the observation that Pns11 overexpression enhances OsSAMS1 activity without up-regulating its mRNA. Additionally, relative to the Pns11 expression level in RDV infection, the transgenically expressed Pns11 level in S11 OX lines was probably low and insufficient to induce significant changes in ethylene pathway genes. Nevertheless, these results indicate that the ethylene pathway genes are regulated by RDV infection, most likely through Pns11-OsSAMS1 interaction. However, additional mechanisms involving other RDV proteins cannot be ruled out at this point.

4) The authors repeatedly cite Satoh et al., (2011) as showing that ethylene related genes are induced following RDV infection (Introduction, Abstract). But that cited paper actually says the opposite: "However, the genes for ET and SA synthesis were not strongly activated by RDV infection." The authors finding that SAMS1 is regulated post-transcriptionally is consistent with the lack of transcriptional changes noted by Satoh et al.,. The rationale for stating that Satoh et al., demonstrate the induction of ethylene related genes needs to be justified (GO analysis of microarray data?) or else interpretation of these earlier results revised and discussed accordingly.

We are sorry for misquoting the published work by Satoh et al., (2011). We have revised the phrase “RDV infection induced expression of ethylene-related genes” into “RDV infection perturbed the expression of several ethylene response genes like ERFs (Ethylene response factors)” in the revised manuscript, please see Introduction and subsection “SAM, ACC, and ethylene contents increased in *S11*and *OsSAMS1*overexpression lines and decreased in *OsSAMS1* CRISPR/Cas9 knockout and RNAi lines”

Satoh et al., (2011) stated “[…] the genes for ET and SA synthesis were not strongly activated by RDV infection”. We also found no strong induction of ET synthesis genes in our own RNA-seq dataset (See Supplementary file 3). Instead, OsSAMS1, the main subject of the current study, is post-transcriptionally modified by Pns11 to become enzymatically more active, leading to increased production of ethylene. However, we did detect a significant transcriptional change in a set of genes that respond to increase ethylene availability (see above).

5) The authors cannot state that virus accumulation 'significantly' increased (subsection “RDV infection”) as they do not provide statistics. Also, they cannot state "the infection rates were much higher in the S11 OX lines compared to Wt (Figure 1)" as the difference is actually rather small, and line OX#3 is the same as Wt.

We apologize for the misleading statements. We have corrected these in the revised manuscript. Please see subsection “Overexpression of Pns11in rice enhances susceptibility to RDV infection”.

6) Supplementary file 2 data suggests that the effect is observed before 2-3 wpi; but at later time points there is no difference between wild type and OX plants. Authors should comment on this; because it is not providing complete resistance to RDV, it is providing tolerance to the RDV suggesting ethylene is not the only factor required for RDV pathogenesis.

Thank you for your comments. We have addressed this issue and discussed in the revised manuscript in the Discussion section as follows:

“We didn’t find a strong difference between WT and *S11* OX#3 (Figure 1), especially in virus accumulation and infection rate. This is probably due to the relatively low expression level of Pns11 in this particular line (Figure 1—figure supplement 1), which may be insufficient to induce a significant increase in ACC and ethylene production (Figure 3). Furthermore, the hyper-susceptibility to RDV in Pns11 overexpressing plants was more prominent prior to 3 wpi (Supplementary file 2). This is easily explained by the fact that enhanced susceptibility allowed more Pns11 transgenic plants to show more conspicuous symptoms at earlier time points. This however does not prevent WT plants from becoming symptomatic at later time points, thus catching up with the transgenic plants in the rate of infected plants.”

7) Figure legends – lack details on timing of tissue collection and days after infection, etc.

Thank you for your comments. We have added details like timing of tissue collection and days after infection etc. in the figure legends.

8) Text in terms resistance should be toned down; it is really tolerance, not resistant. "Inhibits RDV infection" – is misleading because it reduces infection compared to the wild type plants. subsection “Ethylene is induced by viral infection and ethylene accumulation increases host susceptibility”: "RDV infection triggers ethylene synthesis and accumulation through the interaction of Pns11 and OsSAMS1, and resultant activation of OsSAMS1". Although there is indirect data for this conclusion in the paper, there is no direct evidence that this happens during RDV infection.

Thank you for your comments. We have replaced “resistance” with “tolerance”; “inhibits RDV infection” with “reduces infection”.

We have performed a new assay (See subsection “Ethylene is induced by viral infection andethylene accumulation increases host susceptibility” in the revised manuscript) to confirm that Pns11 interacted with OsSAMS1 during RDV infection (See Figure 6—figure supplement 1) but not in RDV-infected Ossams1 KO lines (See Figure 6—figure supplement 2A). We also measured the contents of SAM, ACC and ethylene in RDV-infected Ossams1 KO lines and mock infected Ossams1 KO lines, and found that the RDV-induced increase SAM, ACC and ethylene disappeared in these lines (See Figure 6—figure supplement 2B-D). These data indicated that RDV infection triggers ethylene synthesis and accumulation through the interaction of Pns11 and OsSAMS1 and resultant activation of OsSAMS1.

[Editors' note: further revisions were requested prior to acceptance, as described below.]

You should revise the text (subsection “OsSAMS1 enzymatic activity”). It is misleading to indicate that proteins present in the same fraction means they are interacting. Your BiFC data and Figure 6—figure supplement 1 does show complex formation. Please be sure to carefully revise the text and be sure to note what each experiment really shows.

We apologize for the misleading statements describing the gel-filtration result and have replaced the gel-filtration experiment with IP western experiment (See Figure 2 and subsection “OsSAMS1 enzymatic activity “in the revised manuscript). For the BiFC data, we have conducted all the positive and negative controls to confirm the interaction between Pns11 and OsSAMS1. For the in vivo time course pull-down assay (Figure 6—figure supplement 1), we have chosen one time point experiment (4wpi) to show the interaction between Pns11 and OsSAMS1 during RDV infection (See Figure 2).

Reviewer #2:Authors have addressed most of the issues raised in the last reviews. However, authors fail to respond to this comment:– Figure 2 – This reviewer fail to understand how the authors conclude interaction based on fractionation? This should be a simple IP experiment considering the authors have antibodies to Pns11 and OsSASM1. IP with OsSAMS1 antibody and probe for Pns11, in mock and RDV infected tissues. Alternatively, IP with Pns11 antibody and probe for OsSAMS1.

Thank you for your comments. We indeed have data showing the interaction between Pns11 and OsSAMS1 in RDV-infected rice by in vivo time course pull-down assay using OsSAMS1 antibody. We apologize for the misleading statements and have replaced the gel-filtration experiment with one time point IP western experiment (See Figure 2and subsection “OsSAMS1 enzymatic activity “in the revised manuscript) as described above.